# Patient Outcomes following Immediate Tracheostomy and Emergency Decompressive Craniectomy in the Same Setting

**DOI:** 10.3390/ijerph192315746

**Published:** 2022-11-26

**Authors:** Ramesh Kumar, Mohd Sofan Zenian, Tang Yiu Maeng, Farizal Fadzil, Anis Nabillah Mohd Azli

**Affiliations:** 1Department of Surgery, Faculty of Medicine, Universiti Kebangsaan Malaysia, Kuala Lumpur 56000, Malaysia; 2Department of Neurosurgery, Hospital Queen Elizabeth, Sabah 88300, Malaysia; 3Department of Surgery, Penang General Hospital, Penang 10990, Malaysia

**Keywords:** immediate tracheostomy timing, cranial decompression, mechanical ventilation, critical care, length of stay (LOS)

## Abstract

Early tracheostomy is recommended for patients with severe traumatic brain injury or stroke. Tracheostomy in the same setting as emergency decompressive craniectomy, on the other hand, has never been investigated. Our goal was to compare the outcomes related to the duration of mechanical ventilation in patients who had immediate (IT) vs. early (ET) tracheostomy following an emergency decompressive craniectomy in a Neurosurgical centre in Sabah, Malaysia. We reviewed 135 patients who underwent emergency decompressive craniectomy for traumatic brain injury (TBI) and stroke patients between January 2013 and January 2018 in this retrospective cohort study. The cohort included 49 patients who received immediate tracheostomy (IT), while the control group included 86 patients who received a tracheostomy within 7 days of decompressive surgery (ET). The duration of mechanical ventilation, length of stay (LOS) in the critical-care unit, and intravenous sedation were significantly shorter in the IT group compared to the ET group, according to the study. There was no significant difference between the two groups in the incidence of ventilator-associated pneumonia (VAP), tracheostomy-related complications, or 30-day mortality rate. In conclusion, compared to early tracheostomy, immediate tracheostomy in the same setting as emergency decompressive craniectomy is associated with a shorter duration of mechanical ventilation and LOS in critical-care units with acceptable morbidity and mortality rates. This practise could be used in busy centres with limited resources, such as those where mechanical ventilators, critical-care unit beds, or OT wait times are an issue.

## 1. Introduction

Decompressive craniectomy is a life-saving emergency surgical treatment routinely performed for patients with severe traumatic brain injury and stroke, either malignant middle cerebral artery (MCA) infarction or haemorrhagic stroke [1,2]. Survivors may remain in a minimally conscious state and require prolonged intubation with the assistance of mechanical ventilators to prevent airway obstruction, aspiration, and consequential hypoxia in the intensive-care unit (ICU). The recovery time for these patients is unpredictable and difficult to estimate [3]. Hui et al. and Kalanuria et al. observed that they are more susceptible to ventilator-associated pneumonia (VAP) and have prolonged stays in intensive-care units (ICUs) or high-dependency units (HDUs) [1,3]. VAP in neurosurgical patients may be as high as 40% in mechanically ventilated patients [1,2].

In addition, Hui et al. discovered that the primary reasons for tracheostomy in these TBI patients include inability to be weaned off invasive mechanical ventilation, absence of protective airway reflexes, impairment of respiratory drive, and difficulties managing secretions [2]. The percentage of TBI patients who might benefit from a tracheostomy [3] and the ideal time for the procedure are still unknown, as reported by Elkbuli et al. and Robba et al., and relevant biases confound the limited, mostly retrospective, data on this topic [4,5].

The Brain Trauma Foundation (2020) and stroke studies have recommended tracheostomy over endotracheal intubation to shorten the duration of mechanical ventilation where the overall benefit outweighs the complications associated with such a procedure [5,6,7]. The average day of tracheostomy surgery is conducted 5–7 days following intubation, although the time of early tracheostomy differs between studies. Even though many studies have advocated for early tracheostomy in patients with traumatic brain injury and stroke, tracheostomy performed concurrently with emergency decompressive craniectomy has never been investigated.

The Neurosurgical Department of Hospital Queen Elizabeth (HQE) is the main referral center in the state of Sabah and it is constantly challenged by a lack of ICU, HDU, and ventilator machines. As a result, it is in the patient’s best interests to be weaned off mechanical ventilation earlier to facilitate pulmonary toilet in the general ward. Furthermore, combining tracheostomy with decompression craniectomy in the same setting reduces the need for an additional operating theatre (OT) visit, which is costly. This will also help to alleviate the problem of a crowded OT schedule and the wait for a tracheostomy slot, which may take a few days in our setting. Every additional day that the patient is on the ventilator machine while waiting for a tracheostomy must be considered, as this may put them at risk of VAP [7]

Marra, A. et al. published a systematic review and meta-analysis in 2021 comparing the timing of tracheostomy after decompressive craniectomy. However, none of the studies looked at the effects of tracheostomy and decompressive craniectomy surgery in the same setting. The purpose of this study was to compare the clinical outcomes of immediate tracheostomy versus early tracheostomy in trauma and stroke patients undergoing emergency decompressive craniectomy at HQE’s Neurosurgical Department. The outcomes of this study can ultimately be implemented in other district hospitals in Sabah.

## 2. Materials and Methods

Hospital Queen Elizabeth (HQE) is a tertiary-care facility that serves all district hospitals in the state of Sabah, separated by vast terrains. It has 18 beds in High-Dependency Unit (HDU), mostly for medical patients, and only 4 beds in Neuro HDU, with 16 beds in ICU providing critical care to patients with various illnesses. Traumatic neurosurgical patients will be admitted to the Neuro HDU if they have isolated brain injury or the ICU if they have associated polytraumatic injuries. The neurosurgery team and two nurses trained in neurosurgical patient care manage the hospital’s Neuro HDU. The hospital’s intensive-care unit anesthetist co-manages ventilated patients.

### 2.1. Study Design and Patient Selection

This is a retrospective cohort study of patients who underwent decompressive craniectomy surgery for traumatic brain injury and stroke patients (i.e., malignant MCA infarct and intracerebral bleed) at Hospital Queen Elizabeth in Kota Kinabalu, Sabah, between January 2013 and January 2018.

We identified neurosurgical patients who had undergone emergency decompressive craniectomy surgery. The cohorts consist of patients who received immediate tracheostomy (IT) in the same setting as decompressive craniectomy and the control group will consist of patients who received a tracheostomy within 7 days (ET) of decompression surgery. For IT cohort, tracheostomy and decompressive craniectomy consent were obtained in the same setting. Patients presented with severe TBI, defined by GCS < 8 were subjected for immediate tracheostomy in anticipation for poor GCS recovery after surgery. The inclusion and exclusion criteria will be used to select the cohorts (for a flow chart, see Figure 1).

The following was the inclusion criteria: (a) Adult patient age ≥ 12 years old admitted to HQE. (b) Patients who were treated with supratentorial decompressive craniectomy under emergency list. (c) The group of patients who also received tracheostomy in same setting with decompressive craniectomy or within a week after decompressive surgery.

The exclusion criteria include: (a) Patients < 12 years old. (b) GCS score of 3 or 4 with no spontaneous respiration and bilateral pupillary dilatation. (c) Patients who were extubated successfully following supratentorial decompressive craniectomy (without tracheostomy). (d) Pre-existing tracheostomy. (e) Patients suffering from concurrent life-threatening thoracic trauma. (f) Had been ventilated for >3 days preoperatively. (g) Only required cerebrospinal fluid (CSF) diversion procedures. (h) Required another cranial decompression procedure. (i) Patients with brain tumour, brain abscess and vascular malformation cases. (j) Patients who had posterior fossa decompressive surgery. (k) Patients who died before tracheostomy could be performed.

Incomplete and untraceable patient case notes were also excluded from this study. Considering the ability to conduct decompressive craniectomy in district hospital, we omitted cases such as brain tumor, brain abscess, and vascular malformation from our analysis. While all other Neurosurgical cases are referred to HQE, the general surgeon on staff in the district hospital can perform emergency decompressive craniectomy for patients with severe TBI.

Data regarding baseline clinical characteristics which are the preoperative variables were collected for all patients: age, disease aetiology, preoperative GCS, CT brain findings, time interval between onset of symptoms and emergency surgery, and preoperative APACHE II score.

### 2.2. Operational Definition

Emergency decompressive craniectomy is referred to as craniectomy when the skull vault is removed over a swollen brain and a craniotomy when the bone flap is replaced. Based on the literature on decompressive surgery and tracheostomy, these studies only consider patients who underwent supratentorial decompressive surgery for traumatic brain injury, ischemic stroke, and haemorrhagic stroke [8,9,10,11,12].

All tracheostomies are performed in the operating room using an open-method approach and standardised technique using a low-pressure cuffed tracheostomy tube. Patients were diagnosed with VAP after being intubated for more than 48 h and presenting with clinical examination and radiographic imaging consistent with pneumonia, as well as a positive sputum culture for bacterial growth.

After tracheostomy, weaning off mechanical ventilator was defined as discontinuation of mechanical ventilation or liberation from the mechanical ventilator. When controlled ventilation for intracranial pressure optimization was no longer required, all patients were weaned off the ventilator according to a standardised protocol. This entailed gradually transitioning from continuous mechanical ventilation (CMV) mode to synchronised intermittent mandatory ventilation (SIMV) mode, followed by pressure support (PS), regardless of when the tracheostomy was to be performed. When patients were no longer ventilator dependent and neurologically stable, they were discharged from the ICU or Neuro HDU.

The World Federation of Neurological Surgeons’ Glasgow Outcome Score GOS was used to assess the functional outcome. The following neurological outcomes were assessed: 1 = death; 2 = persistent vegetative state with inability to interact with the environment; 3 = severe disability with inability to live independently, but with the ability to follow commands; 4 = moderate disability with the ability to live independently, but with the inability to return to work or school; 5 = mild or no disability with the ability to return to work or school (favourable outcome: 3,4,5) (unfavourable outcome: 1,2)

### 2.3. Statistical Analysis

The analysis was performed using SPSS for Windows version 16.0 (SPSS Inc., Chicago, IL, USA). For comparing the means of two normally distributed quantitative datasets, the student *t*-test was used. The Mann–Whitney U-test was used to compare the median with the first and third (Q1–Q3) quartiles for two abnormally distributed quantitative datasets. The Chi square test was used to compare qualitative variables and the results were expressed as percentages. Statistical significance was defined as a probability value of *p* 0.005.

### 2.4. Ethical Consideration

This research was registered in accordance with the protocol with the National Medical Research Register (NMRR-18-660-40989). Prior to the start of data collection, permission was obtained from the Malaysian Research Ethics Committee (KKM/NIHSEC/P18-832(7)). This study was also approved by the Research and Ethical Committee of UKM Medical Centre (Reference no: UKM PPI/111/8/JEP-2018-303).

## 3. Results

We recruited 386 patients who underwent decompressive craniectomy at HQE for this five-year evaluation. The study omitted patient case notes that were incomplete or untraceable. While 103 patients underwent immediate tracheostomy (IT), 283 patients underwent emergency decompressive craniectomy followed by ET within a week. However, only a total of 135 patients met the inclusion criteria. The IT group had 49 patients, while the ET group had 86 patients. We excluded situations, such as brain tumour, brain abscess, and vascular malformation, from our research because district hospitals are only equipped to conduct decompressive craniectomy involving traumatic brain injury. The district hospital’s on-staff general surgeon can perform an emergency decompressive craniectomy for patients with severe TBI, while all other neurosurgical cases were referred to HQE.

The initial cerebral insult in IT was more severe, as evidenced by the GCS on presentation. The mean GCS on arrival at the emergency department was substantially lower for IT than the ET, 7.6 vs. 9.1 and *p*-value 0.02. According to the CT brain findings in trauma and stroke patients, the IT group had more severe brain injuries than ET patients, as depicted in Table 1. Despite ET having a larger patient population, the IT group has a higher percentage of patients who have subdural haemorrhage (100%), subarachnoid haemorrhage (26.3%), intraventricular haemorrhage (10.5%), and intracerebral haemorrhage (84.8%). Because of this, immediate urgent tracheostomy was chosen for this patient population.

Apart from that, age, disease aetiology, pupil reactivity, pre-operative GCS, co-morbid illness, the time between the onset of symptoms and the emergency operation, the time between the neurosurgical assessment and the emergency operation, the pre-operative APACHE II score, and the length of the operation did not significantly differ between the two groups, with a *p*-value ranging from 0.1 to 0.3.

The mean time from disease onset to emergency procedure did not statistically differ between the two groups. In contrast, both groups took longer than other studies to perform an emergency decompressive craniectomy operation because of Sabah’s terrain and geographic organization, which indirectly increased the distance between district hospitals and Hospital Queen Elizabeth (HQE).

What stands out in Table 1 is the time interval between development of disease and placement of tracheostomy was dramatically reduced in IT compared to ET, with mean values of 11.1 (SD 4.5) and 86.0 (SD 28.2) hours, *p*-value < 0.001, respectively.

Table 2 summarizes the outcomes for the patients in both groups. It is apparent from this table that mechanical ventilation duration (72.0 (IQR 30.0) vs. 120.0 (IQR 66.0) hours, *p* < 0.001), LOS in Neuro HDU/ICU (96.0 (IQR 78.0) vs. 180.0 (IQR 72.0) hours, *p* < 0.001), and duration of intravenous sedation (48.0 (IQR 30.0) vs. 96.0 (IQR 68.3) hours, *p* < 0.002) were significantly shorter in the IT than ET group. These results can be found in Figure 2 and Figure 3. There was no significant difference between the two groups in the time from tracheostomy to ventilator weaning, the LOS in the neurosurgical ward, the incidence of VAP, the proportion of tracheostomy-related complications, or the 30-day mortality rate.

Table 3 illustrates that there was no significant difference between the two groups’ Glasgow Outcome Scores (GOS) at discharge or six months. According to Table 4, the proportion of trauma to stroke patients who underwent decompressive craniectomy surgery was 46.7% and 53.3%, respectively. There were no appreciable changes between the two groups, despite the higher distinction of stroke patients who underwent IT (51.2%) compared to ET (48.8%). Figure 4, on the other hand, summarizes the overall proportion of tracheostomy-related complications.

The single most striking observation to emerge from the data comparison was that, in a subgroup analysis of trauma and stroke patients, immediate tracheostomy significantly reduces the duration of mechanical ventilation from intubation to ventilator weaning and Neuro HDU/ICU LOS (Figure 5 and Figure 6).

## 4. Discussion

There has been discussion on when tracheostomy should be performed on neurosurgical patients. Prior studies by Elkbuli et al. and Robba et al., where the sample population was heterogeneous, showed that tracheostomy provided benefits in the critical-care situation, such as shorter duration of mechanical breathing and LOS in ICU and HDU [4,5,6,7,8,12]. Only few studies looked at the timing of tracheostomy in patients who have had decompressive craniectomy [4,5,12,13,14,15]. The earlier the weaning period, in terms of mechanical ventilation and HDU or ICU stay, the sooner the post-operative patient can be moved to the general ward for ongoing care. According to McCredie et al. [15], this will boost the turnover rate for the mechanical ventilators and beds, both of which are badly needed but scarce.

We investigated the effects of tracheostomy performed immediately following decompressive craniectomy on trauma and stroke patients. From the study, we discovered that the duration of mechanical ventilation, LOS in Neuro HDU/ICU as well as duration of intravenous sedation can be reduced while maintaining acceptable mortality and morbidity rates. These results are comparable to a recent systematic review and meta-analysis conducted by Marra et al. in 2021 [13].

According to a literature review, 35–48% of neurosurgical patients with GCS 7 or less will eventually require tracheostomy for prolonged ventilation [7,9,16,17]. Goettler et al. [11] predicted that 80% of trauma patients with GCS less than 5 who underwent emergency surgery would require tracheostomy, whereas Kim et al. [9] focused on patients who underwent craniectomy for traumatic brain injury and found that patients with a GCS score of 8 are unlikely to be weaned off mechanical ventilation. On the other hand, Catalino et al. [8] found a 56–70% risk of VAP in patients on endotracheal tube intubation. Hence, in HQE, neurosurgical patients with preoperative GCS less than 8 will need tracheostomy, either in the same setting or in the early post-operative period.

Based on our research, the neurosurgeon who is “on call” will determine when tracheostomy will be performed in the event where a patient is scheduled for an emergency decompressive craniectomy. The patient’s clinical parameters and availability of hospital resources were taken into consideration when making these decisions. Patients with GCS < 8 were subjected to immediate tracheostomy rather than an early one since they had more serious brain damage, bleeding, and stroke. The scarcity of critical-care beds also played a role at this juncture.

Choosing the ideal time to perform a tracheostomy is difficult. The lack of a standardized process in deciding which patients will benefit from tracheostomy makes things even worse. Early tracheostomy appears advantageous in individuals with severe trauma TBI or stroke. Our study would benefit greatly from the use of the stroke-related early tracheostomy score (SETscore) to aid in the objective selection of participants for IT and ET.

The utilization of a tracheostomy in patients requiring neurocritical care can be determined using the internal screening tool, SETscore [18]. The score is performed within the first 24 h after admission. Neurological function, neurological lesion, and general organ function/procedure are among the criteria used in the SETscore [19]. SETscore >10 denotes early tracheostomy candidacy. Isabel et al. applied the usage of SETscore to intracranial bleed and stroke patients with and without trauma, which is a great example of how this can be adapted in our study [18].

Earlier tracheostomy helped shorten the duration of mechanical ventilation, ultimately improving oxygenation to the brain when compared to endotracheal intubation [7,10]. Effective decompressive craniectomy surgery reduces intracranial pressure (ICP), enhances cerebral blood flow, and may minimize ischemic damage by improving oxygenation of the brain tissue [6,11]. ICP can, on rare occasions, continue to increase within two days of operation. Therefore, the goal is to break this cycle, since earlier tracheostomy results in shorter duration of mechanical ventilation and better brain oxygenation as compared to endotracheal intubation [6,20]. Tracheostomy also allows for deeper suctioning of airway secretions, which aids in pulmonary toilet and oral hygiene [8,17,21,22,23] and, subsequently, enabled patients to be nursed for oral feeding and early ambulation [16,17,24].

According to Arabi et al. and Shibahashi et al., the expected benefits of shorter mechanical ventilation and LOS in Neuro HDU/ICU included more efficient healthcare allocation and medical cost savings [10,16,25]. In the state of Sabah, even a tertiary-care centre like ours is constantly short of Neuro HDU/ICU beds. Therefore, combining tracheostomy with decompressive craniectomy helped to address the critical-care unit’s bed scarcity as well as their restricted supply of ventilators. Furthermore, an immediate tracheostomy helped to reduce the need for additional operating theatre (OT) visits, which are costly [10,11]. This could indirectly alleviate the problem of a hectic OT schedule, as the wait for an available tracheostomy OT slot may take a few days, which put the patients at risk of VAP [5,7,10]. The timing measured in our study was in hours instead of days because patient movement within the hospital was dynamic and subject to the availability of ventilator machines and beds in Neuro HDU/ICU. 

Apart from that, the VAP rates were not significantly different between the two groups. We believed this was because the IT group had a higher proportion of stroke patients (61.2%). It is hypothesized that stroke patients have a compromised airway immune system, varying degrees of respiratory failure, and a risk of micro-aspiration at the time of stroke onset, making them more susceptible to infection [8,21,22]. Cheung et al. observed that during endotracheal intubation, contaminated oropharyngeal secretions may be aspirated into the lung, eventually forming a bacterial biofilm on the surface of the endotracheal tube [17,21]. This contributes to VAP development because biofilm fragments are dislodged and blown into the lung by ventilator gas flow [21].

The negative effects of tracheostomy should also be considered. However, the effects of the timing of tracheostomy on mortality are still controversial [12]. In our investigation, we discovered a non-significant reduction in 30-day mortality in the IT group. The overall mortality of both groups in our study was equivalent to the 3–8% rate seen in the early tracheostomy group (performed within 7 days) for patients who underwent emergency decompressive craniectomy surgery [7,15,20].

## 5. Limitations

There are a few limitations in our study. The decision of when to perform tracheostomy may involve consideration of multiple factors, including patient characteristics, severity of illness, physician judgment, availability of resources, expected clinical course, and difficulty in obtaining consent from the patients’ families.

Our retrospective cohort study is susceptible to bias. However, the bias between the two groups was minimized because there were no significant differences in the preoperative variables between the two groups. This bias could also be avoided by using SETscore before tracheostomy timing decisions are made. The perception of the severity of the developing cerebral oedema and a mass effect potentially influence decision making. Patients with lower degrees of secondary brain injury may have been selected for the early tracheostomy group as compared to immediate tracheostomy.

We also faced difficulty obtaining consent from a patient’s next of kin when they were intubated, which may affect the timing of tracheostomy. The consent for the IT group was obtained concurrently with the consent for cranial surgery. Another drawback was that the time interval between symptoms and surgery was longer (13 h), as opposed to the standard 4–6 h recommended for cranial decompression [12,20,26]. The major cause was attributed to Sabah’s vast geographical terrain, where an ambulance journey from one district to our neurosurgical centre in Kota Kinabalu typically took 3–4 h.

Another factor to consider was the availability of critical-unit beds and mechanical ventilation in ICU or Neuro HDU. Consequently, the intubated patients will only be admitted to the operating room for emergency decompressive craniectomy once a mechanical ventilator is available. Finally, we did not comment on tracheostomy decannulation rates because very few patients had their tracheostomy tubes removed completely in the hospital. Patients will be seen at their respective district hospitals to continue their rehabilitation process. There, medical personnel are trained in tracheostomy tube care and subsequent decannulation. Thus, many of these post-operative patients are tracheostomy free during their neurosurgical clinic review at HQE.

Given the shorter length of stay in the ICU and Neuro HDU, there are likely to be economic benefits and better resource utilization from the immediate tracheostomy group, which was not explicitly studied here. Prospective controlled studies are warranted to provide solid evidence, especially in terms of cost effectiveness for the true benefits of immediate tracheostomy following emergency decompressive craniectomy.

## 6. Conclusions

In conclusion, the findings of this study suggested that immediate tracheostomy after decompressive craniectomy surgery may lessen mechanical ventilation duration and LOS in the critical-care unit. The combined procedures in the same setting could be more beneficial than the procedure’s anticipated complications. This practise could be used in low-resource but busy district hospitals, where mechanical ventilators, critical-care unit beds, or OT waiting times are a problem. The findings of this study can be used to plan future studies involving tracheostomy and decompressive surgery, in terms of a cost–benefit analysis.

## Figures and Tables

**Figure 1 ijerph-19-15746-f001:**
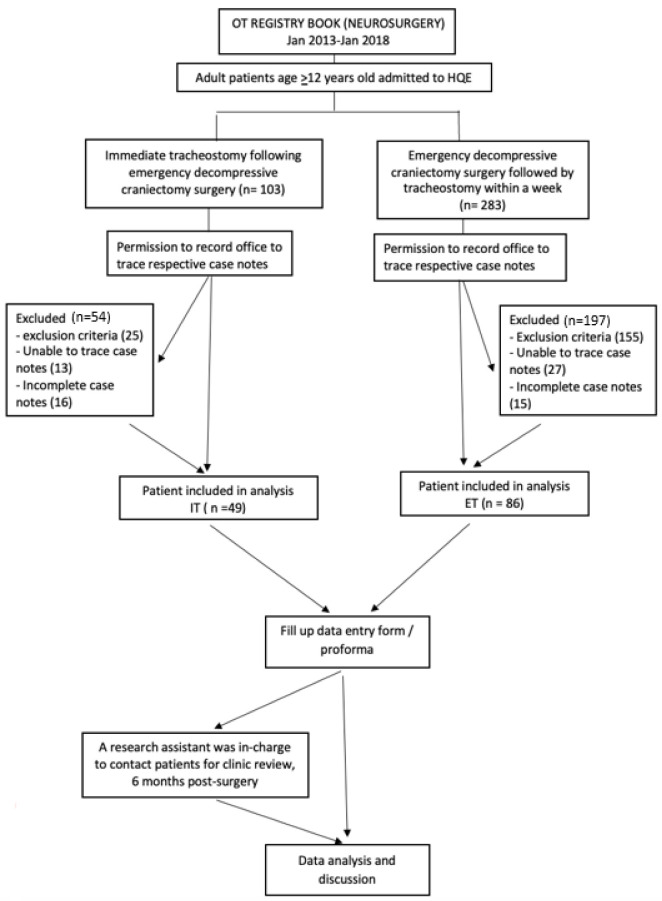
Flow chart.

**Figure 2 ijerph-19-15746-f002:**
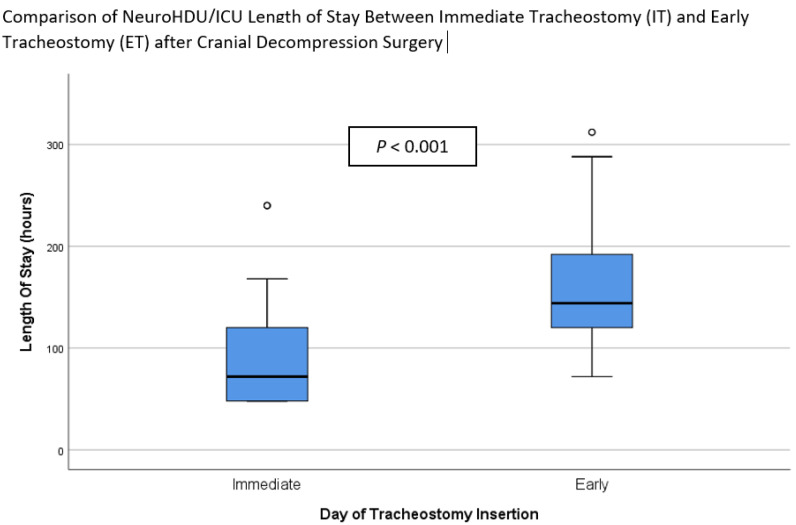
Comparison of Neuro HDU/ICU Length of Stay (LOS) between Immediate Tracheostomy (IT) and Early Tracheostomy (ET) after decompressive craniectomy surgery.

**Figure 3 ijerph-19-15746-f003:**
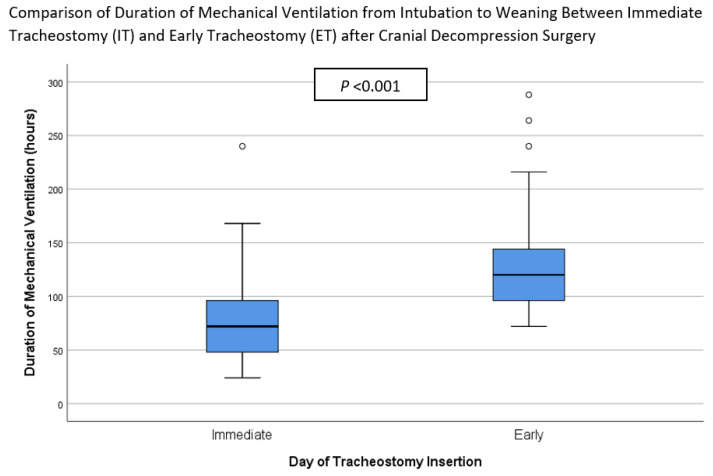
Comparison of duration of mechanical ventilation from intubation to weaning between Immediate Tracheostomy (IT) and Early Tracheostomy (ET) after decompressive craniectomy surgery.

**Figure 4 ijerph-19-15746-f004:**
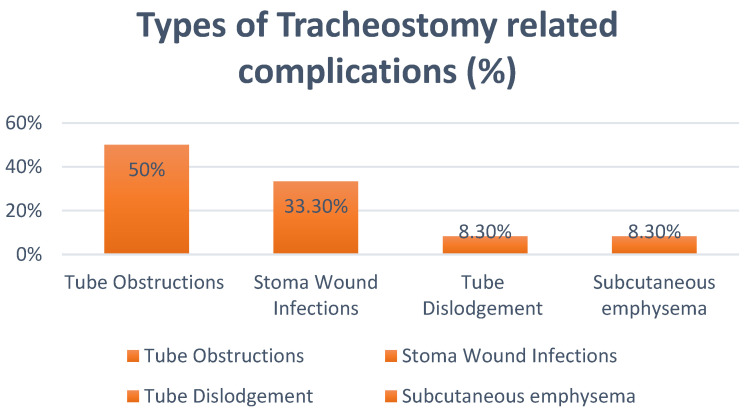
Types of tracheostomy-related complications.

**Figure 5 ijerph-19-15746-f005:**
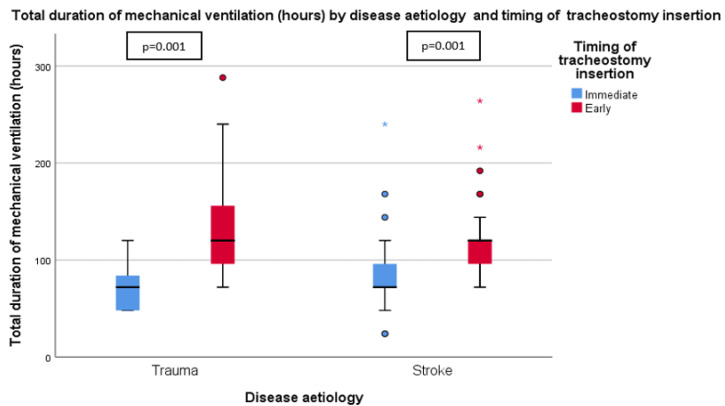
Total duration of mechanical ventilation (hours) by disease aetiology and timing of tracheostomy insertion.

**Figure 6 ijerph-19-15746-f006:**
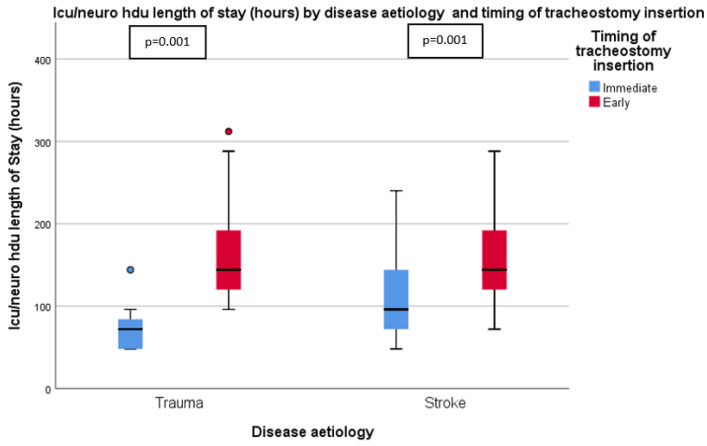
ICU/Neuro HDU length of stay (hours) by disease aetiology and timing of tracheostomy insertion.

**Table 1 ijerph-19-15746-t001:** Comparison of baseline clinical characteristics between Immediate Tracheostomy (IT) and Early Tracheostomy (ET) after decompressive craniectomy surgery.

Clinical Characteristics	Total N = 135 (%)	IT N = 49 (%)	ET N = 86 (%)	*p*-Value
**Mean Age in years** **(SD)**	48.5 (15.0)	50.0 (13.8)	47.6 (15.7)	0.374 ^#^
**Age Groups**				0.134 *
<40	49 (36.3)	13 (26.5)	36 (41.9)	
41–60	60 (44.4)	27 (55.1)	33 (38.3)	
>60	26 (19.3)	9 (18.4)	17 (19.8)	
**Ethnicity**				0.693 *
KDM	69 (51.1)	27 (55.1)	42 (48.8)	
Chinese	15 (11.1)	5 (10.2)	10 (11.6)	
Bajau	14 (10.4)	5 (10.2)	9 (10.5)	
Malay/Brunei	18 (13.3)	4 (8.2)	14 (16.3)	
Others	19 (14.1)	8 (16.3)	11 (12.8)	
**Disease etiology**				0.165 *
Trauma	63 (46.7)	19 (38.8)	44 (51.2)	
Stroke	72 (53.3)	30 (61.2)	42 (48.8)	
**Pupil reactivity at admission**				0.369 *
Normal	73 (54.1)	29 (59.2)	44 (51.2)	
Abnormal (anisocoric/fixed)	62 (45.9)	20 (40.8)	42 (48.8)	
**CT brain findings of Trauma patients**	**Total** **N = 63 (%)**	**IT** **N = 19 (%)**	**ET** **N = 44 (%)**	
Subdural haemorrhage	57 (90.5)	19 (100.0)	38 (86.4)	0.166 *
Subarachnoid haemorrhage	11 (17.5)	5 (26.3)	6 (13.6)	0.224 *
Intraventricular haemorrhage	2 (3.2)	2 (10.5)	0 (0.0)	0.088 *
Contusional haemorrhage	35 (55.6)	10 (52.6)	25 (56.8)	0.759 *
Epidural haemorrhage	13 (20.6)	2 (10.5)	11 (25.0)	0.311 *
Skull fracture	39 (61.9)	13 (68.4)	26 (59.1)	0.484 *
**CT brain findings of Stroke patients**	**Total** **N = 72 (%)**	**IT** **N = 30 (%)**	**ET** **N = 42 (%)**	0.086 *
Major ischemic infarct	11 (15.2)	2 (6.7)	9 (21.4)	
ICH	61 (84.8)	28 (93.3)	33 (78.6)	
**Status of basal cisterns**	**Total** **N = 135 (%)**	**IT** **N = 49 (%)**	**ET** **N = 86 (%)**	0.823 *
Normal	12 (8.9)	4 (8.2)	8 (9.3)	
Abnormal	123 (91.1)	45 (91.8)	78 (90.7)	
**Midline shift**				0.301 *
Yes	123 (91.1)	43 (87.8)	80 (93.0)	
No	12 (8.9)	6 (12.2)	6 (7.0)	
**Mean GCS on arrival to emergency department** **(SD)**	8.6 (2.9)	7.6 (3.1)	9.1 (2.7)	0.002 ^#^
**Mean GCS pre-op** **(SD)**	6.0 (1.3)	5.9 (1.1)	6.0 (1.4)	0.622 ^#^
**Co-morbid illness**				0.509 *
0	63 (46.7)	21 (42.9)	42 (48.8)	
1	44 (32.6)	19 (38.8)	25 (29.1)	
>2	28 (20.7)	9 (18.3)	19 (22.1)	
Mean time interval from onset of symptoms to emergency operation (hours) (SD)	13.8 (9.3)	14.8 (10.1)	13.4 (8.9)	0.414 ^#^
Mean time interval from assessment to emergency operation (hours) (SD)	4.0 (1.9)	4.3 (1.8)	3.9 (1.9)	0.168 ^#^
Mean Pre-operative APACHE II score (%) (SD)	29.9 (12.6)	30.8 (15.2)	29.4 (10.9)	0.544 ^#^
Mean time interval from disease onset to Insertion of tracheostomy (hours) (SD)	58.8 (42.6)	11.1 (4.5)	86.0 (28.2)	**<0.001** ^#^
Duration of operation (hours) (Median -IQR)	3.3 (1.1)	4.0 (1.0)	3.0 (1.2)	0.780

* Chi Square test. ^#^ Student *t*-test.

**Table 2 ijerph-19-15746-t002:** Outcomes of patients who underwent Immediate Tracheostomy (IT) and Early Tracheostomy (ET) after decompressive craniectomy surgery.

Outcomes	Total (N = 135)	IT (N = 49)	ET (N = 86)	*p*-Value
Duration of mechanical ventilation from intubation to ventilator weaning (hours) (Median–IQR)	107.9 (45.9)	72.0 (30.0)	120.0 (66.0)	**<0.001** ^ύ^
Time from tracheostomy to ventilator weaning (hours) (Median –IQR)	50.1 (35.4)	48.0 (24.0)	60.0 (48.0)	0.180 ^ύ^
Duration of Neuro HDU/ICU length of stay (LOS) in (hours) (Median–IQR)	135.8 (60.2)	96.0 (78.0)	180.0 (72.0)	**<0.001** ^ύ^
Duration of intravenous sedation administration while on mechanical ventilation (hours) (Median–IQR)	73.5 (42.8)	48.0 (30.0)	96.0 (68.3)	**0.002** ^ύ^
Duration of neurosurgical ward Length of stay (LOS) (days) (Median–IQR)	17.0 (4.8)	17.0 (4.8)	16.5 (5.0)	0.304 ^ύ^
Incidence of VAP (%)	62 (45.9)	26 (53.1)	36 (41.9)	0.209 *
Proportion of tracheostomy related complications (%)	12 (8.9)	4(4.4)	8 (7.6)	0.823 *

^ύ^ Mann-Whitney U test; * Chi Square test.

**Table 3 ijerph-19-15746-t003:** Glasgow Outcome Score (GOS) of patients who underwent Immediate Tracheostomy (IT) and Early Tracheostomy (ET) after decompressive craniectomy surgery.

At the Time of Discharge	Total N = 135 (%)	IT N = 49 (%)	ET N = 86 (%)	*p*-Value
Favourable (3–5)	121 (89.6)	43 (87.7)	78 (90.7)	0.590 *
Unfavourable (1–2)	14 (10.4)	6 (12.3)	8 (9.3)	
Mortality within 30 days	6 (4.4)	3(6.5)	3 (3.5)	0.475 *
At 6 months (%)	Total N = 129 (%)	IT N = 46 (%)	ET N = 83 (%)	*p*-value
Favourable (3–5)	125 (96.9)	44 (95.7)	81 (97.6)	0.543 *
Unfavourable (1–2)	4 (3.1)	2 (4.3)	2(2.4)	

* Chi Square test.

**Table 4 ijerph-19-15746-t004:** Timing of tracheostomy by disease aetiology.

Disease Aetiology	Total N = 135 (%)	IT N = 49 (%)	ET N = 86 (%)	*p*-Value
Trauma	63 (46.7)	19 (38.8)	44 (51.2)	0.165 *
Stroke	72 (53.3)	30 (61.2)	42 (48.8)	

* Chi Square test.

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
