# Peer review of "Patient Outcomes following Immediate Tracheostomy and Emergency Decompressive Craniectomy in the Same Setting"

_ijerph, 2022, doi:10.3390/ijerph192315746_

Round 1

Reviewer 1 Report

The authors used a retrospective cohort receiving craniectomy to compare outcome factors between IT and ET. They concluded that IT could reduce the patient's consumption of medical resources (duration of mechanical ventilation and LOS in critical care units).

1.     P.2, L.78: The tracheostomy is an invasive treatment and carries risks of different complications. For the patients receiving craniectomy in this study, they were extremely impossible to sign the consent before surgery. Additionally, not all patients receiving craniectomy need tracheostomy (P.3, L. 96-97). Therefore, although the authors used retrospective data, the reviewer expects them to clarify the indication for IT in this manuscript, meaning what kind of patients need to receive concurrent craniectomy and tracheostomy.

2.     In this study, must consent for IT be obtained before the surgery?

3.     The content of the Flow chart is different from the inclusion criteria mentioned (p.3, L.93-94). The author is required to modify the flow chart according to the inclusion criteria.

4.     P.3, L.101-102: Please explain why patients with a brain tumor, brain abscess, and vascular malformation were excluded from this study.

5.     P.4, L.150-166: The results should be shown in a table format to present the results clearly.

6.     P.6, L.184-188: This text is better presented in a table format so that the reader can completely read the results of the comparisons.

Reviewer 2 Report

This paper “Patient outcomes following immediate tracheostomy and emergency decompressive craniectomy in the same setting” includes interesting topics in critical care and is well written. Authors compares two groups; patients with immediate tracheostomy and those received tracheostomy within 7 days after decompressive craniectomy. Shorter LOS, sedation, and mechanical ventilation were found. However, many concerns are still remained before accepting.

Major revision 

1 There is a huge selection bias when patients are allocated in each group. Authors states GCS in IT group was lower than ET group, which affect the physicians’ discretion. We understand this is a retrospective cohort study, however, this bias influenced the results obtained from this study. Authors must omit this huge bias.  

2 Table of background characteristics is missing. 

3 Whether the patients with craniectomy can extubate depends on injured location of brain, severity, amount of hemorrhage, etc., which were not involved in the analysis.

4 Many studies reported the advantage and disadvantage of early tracheostomy among patients with not only traumatic brain injury, but also critical illness, which is consistent with the results of this study. Shorter duration of sedative drug, ventilation free day, and ICU LOS are reported already. Unfortunately, this reviewer did not find anything new in this study. 

5 There is no concrete results and discussion about the cost. Authors describe the benefit of early tracheostomy in busy hospital and availability of critical care unit including neuro HDU. However, there might be some patients who could avoid tracheostomy in natural course among IT group. If authors put the emphasis on the cost-effective, detailed information is necessary. 

6 Deepen the discussion with the optimal score for possibility of extubating such as SET score (Neurocrit Care (2019) 30:185–192).  

Minor 

Figure2 is duplicated.  

Figure4 is missing.

There are some typo and inappropriate position for comma/period, proofread again.

IQR should be displayed with 25%-75%.

Round 2

Reviewer 2 Report

Author answers inquiries from reviewers sincerely. This paper revised sophistically and is worth publishing in IJERPH.

Some minor revision/typo are remained and require modification.

Minor revision

Display the number of patients excluded in Figure 1 in detail (ex. GS3/4 n=..., ). 

Authors displayed characteristics in table1, however, some findings confuse readers. For instance, percentage was calculated with using total number (IT 49 and ET 86), but subdural hemorrhage was 19 (100%). Authors used  percentage with mixing total number and subgroup, respectively, which will make readers confused. 

Table 4 demonstrate "timing of tracheostomy by disease aetiology" right? what is units?

Typo  

P2, line60 "wad"

"hdu" and "dtiming" in Figure 6.

Author Response

First of all, thank you for your response.

Response to the review : 

  1. The typo has been edited
  2. Number of patients excluded has been included in Figure 1
  3. The total numbers are represented by and the percentage are in the bracket (%).